# Moral decision-making is altered in patients with schizophrenia

**Koohyar Ahmadzadeh[1]☯, Atiye Sarabi-Jamab[2]☯, Farahnaz Beiranvand[3], Mina Forouzandeh[4], Fatemeh Sadat Mirfazeli[5,6,7]\*‡, Seyed Vahid Shariat[7]\*‡**

1 Physiology Research Center, Iran University of Medical Sciences, Tehran, Iran, 2 Faculty of Governance, University of Tehran, Tehran, Iran, 3 Mental Health Research Center, Iran University of Medical Sciences, Tehran, Iran, 4 Department of Medical Ethics, School of Medicine, Iran University of Medical Sciences, Tehran, Iran, 5 National Brain Centre, Iran University of Medical Sciences, Tehran, Iran, 6 Faculty of Advanced Technologies in Medicine, Iran University of Medical Sciences, Tehran, Iran, 7 Mental Health Research Center, Psychosocial Health Research Institute, Department of Psychiatry, School of Medicine, Iran University of Medical Sciences, Tehran, Iran

☯ These authors contributed equally to this work and share first authorship
‡ FSM and SVS also contributed equally to this work and share corresponding authorship.
\* Vahid.shariat@gmail.com (SVS); Mirfazeli.f@iums.ac.ir (FSM)

## Abstract

### Introduction

Previous research suggests patients with schizophrenia have altered decision-making when presented with philosophical moral scenarios. However, not much is known about everyday moral decision-making in patients, which can be more relevant to their real-life social functioning.

### Methods

32 patients with schizophrenia and 32 control subjects were investigated using everyday moral vignettes. The vignettes consisted of two moral and immoral categories. Participants were asked to rate each vignette in dimensions such as emotional intensity, emotional aversion, harm, self-benefit, other-benefit, pre-mediation, legality, social norm violation, and moral appropriateness.

### Results

Patients with schizophrenia evaluated immoral vignettes as less morally inappropriate compared to control subjects ($F(1,62) = 15.5$, $p = 0.0002$). Evaluation of benefit was also different between patients and control subjects, with patients evaluating self-benefit to be higher in moral vignettes ($F(1,62) = 9.167$, $p = 0.004$), and other-benefit to be higher in immoral vignettes ($F(1,62) = 9.7$, $p = 0.003$). No significant difference was observed in the assessment of emotional intensity and the remaining dimensions.

**Data availability statement:** Due to the policy of the responsible ethics committee, we do not have permission to make single-subject data publicly available without participants direct consent. The used datasets are available from the National Brain Centre upon personal reasonable request, which can be made via email (bic@iums.ac.ir). The centre director will grant access to the used datasets after consulting with and approval of the Iran University of Medical Sciences ethics committee.

**Funding:** The author(s) received no specific funding for this work.

**Competing interests:** The authors have declared that no competing interests exist.

## Conclusion

Most aspects of everyday moral decision-making remain intact in patients with schizophrenia. The observed deficits appear to be limited to the evaluation of moral appropriateness in immoral situations and overall benefit assessment. The distortion in benefit assessment for oneself and others could be a crucial area for targeted intervention in patients with schizophrenia.

## Introduction

Patients with schizophrenia have altered social decision-making capabilities, which may be partially attributed to difficulties in accurately understanding social cues and contexts and indifference toward finding a fair solution [1–3]. These patients show poorer performance on tests such as Ultimatum Game (UG) and Dictator Game; however, the extent of impairments varies depending on the employed measures [4,5].

Moral decision-making involves the ability to assess complex situations, consider multiple aspects, and make judgments based on understanding what is morally acceptable or not [6,7]. As moral judgment often relies on an individual's ability to understand and consider the perspectives of others, it is possible that impairment in social cognition may contribute to difficulties in moral decision-making [8]. Most studies investigating moral decision-making in patients with schizophrenia have employed philosophical moral vignettes such as Tsedek Test and Kohlberg's moral judgment interview. In these studies, patients with schizophrenia had more interest in personal gain, were indifferent to harm to others, and scored lower in Kolhbergs moral judgment interview [3,6,9].

Previous research has demonstrated that individuals employ different reasoning strategies when presented with everyday moral situations as opposed to complex unconventional philosophical scenarios [6,10,11]. These findings underscore the importance of using everyday moral decision-making as they may elicit more natural and relevant responses, thereby providing a more accurate understanding of moral reasoning abilities in patients with schizophrenia.

Moreover, the focus on everyday moral dilemmas with multi-aspect dimensions could provide precious insights into the functional implications of these deficits, ultimately contributing to the development of targeted interventions such as socio-cognitive rehabilitation to enhance social functioning in patients with schizophrenia [12].

This study investigates everyday moral decision-making in patients with schizophrenia and demonstrates that most aspects of moral decision-making remain intact in these patients. The observed differences seem to be limited to benefit assessment and evaluation of moral appropriateness.

## Materials and methods

### Participants

This cross-sectional study recruited 64 participants (32 patients, 32 control subjects) from clinics of university-affiliated hospitals between June 1st and November 30th of 2021.

Patients from two different outpatient psychiatry clinics who were diagnosed with schizophrenia according to the Diagnostic and Statistical Manual of Mental Disorders (DSM)-5, as established by an attending psychiatrist, were included in the study. Patients in the acute phase of the disease, experiencing severe symptoms that hindered their ability to answer the questionnaires, and those with changes in medications within the past month were excluded from the study.

Control subjects were recruited from all outpatient clinics of university-affiliated hospitals, excluding psychiatry, psychology, neurology, and neurosurgery clinics. Subjects with a General Health Questionnaire (GHQ-12) score of less than 23 were included as the control group. Individuals with a history of mental illness or psychiatric hospital admission, either personally or among their first-degree relatives, were excluded from the control group.

In addition, all subjects younger than 18 or older than 65 years, with an education level less than a high school diploma, a history of head trauma, seizures, cancer, brain surgery, a neurological condition affecting cognitive capabilities, substance use in the past month, alcohol use disorder or alcohol use in the past 48 hours, and history of participating in similar research were excluded from the study. Alcohol use disorder was assessed through interview based on DSM-5 criteria. There were no dropouts throughout the study, and participants responded to all items in the questionnaire.

## Measures

We used an abbreviated paper-and-pencil version of the moral questionnaire originally developed by Escobedo [13] and condensed and standardized by Knutson et al. [11] in this study. The original set of vignettes contains 150 short everyday first-person vignettes categorized as either moral or immoral. Participants were asked to rate each vignette based on 13 dimensions of emotional intensity, emotional aversion, harm, self-benefit, other-benefit, pre-mediation, legality, social norm violation, socialness, frequency, personal familiarity, general familiarity, and moral judgment using a seven-point Likert scale. For example, two of the vignettes are "During my commute through downtown, I see a lot of homeless people. One day I was driving and saw a homeless woman walking her dog. I pulled over and gave her some money" for the moral category and "I was visiting with my grandmother last weekend. I became really tired of talking with her and I wanted to leave. So I told her that I needed to go to work even though I didn't need to" for the immoral category. Due to the length of the original questionnaire, a shorter version, containing 40 vignettes, has been translated, abbreviated, and validated in the Persian language by Yazdanpanah et al [14], which was utilized in this study. The internal consistency of the shortened vignettes in the Iranian culture, measured by Cronbach's alpha, was 0.895. The Cronbach's alpha measurement of the original 150-vignette questionnaire was 0.96 and it was shown to be consistent with Iranian culture [14].

The mental health of control subjects was measured by a Persian-translated and validated version of GHQ-12. This questionnaire is considered valid and reliable across various settings and cultures, such as Iran (Cronbach's alpha measure: 0.87, Convergent validity: r = -0.56, P < 0.0001) [15], Finland (Cronbach's alpha measure: 0.92), India (Cronbach's alpha measure: 0.82), and South Korea (Cronbach's alpha measure: 0.81) [16].

## Ethical considerations

Written informed consent was obtained from the participants and/or their guardians. The capacity to provide consent was determined through the clinical judgment of the attending physician, ensuring participants understood the study's purpose and their voluntary participation. The patients who had the capacity for consent were included in the study. The ethics committee of the Iran University of Medical Sciences provided approval for all study procedures (IR.IUMS.REC.1400.079). All study procedures adhered to the latest version of the Helsinky Regulation.

## Statistical analysis

Descriptive statistics, chi-squared test, and independent sample t-test were used to analyse the data. The significance level was considered as 0.05. Data with normal distribution are presented as mean ± SD, otherwise, data and demographic characteristics are reported as median (IQR) and median [range].

One-way ANOVA test was used to analyze responses to the questionnaire and their relationship to other variables and check for statistically significant differences between the two groups based on participants' average responses. The averages of participants' responses to each dimension were compared between groups based on two categories (moral and immoral). As multiple hypotheses were tested simultaneously, the most possible type I errors were considered as the 13 dimensions in the questionnaire. Accordingly, the significance level of ANOVA test was set as 0.0038. The Bonferroni adjustment was used to correct the experiment-wise error rate. All statistical analyses were performed using R (version 3.6.1; R Core Team, 2019] in RStudio (RStudio- 1.2.5001).

## Results

### Demographic characteristics

32 patients with schizophrenia and 32 control subjects with an overall median age of 37.5 years were included in the study. There were no group differences in age, gender, and education level. The median disease duration among patients was 11 [1–39] years, with 1.5 [0–15] numbers of hospitalization (Table 1).

### Moral vignettes

In moral vignettes, patients with schizophrenia rated self-benefit higher than the control group ($F(1,62) = 9.167$, $p = 0.004$). No significant difference was observed between patients and control subjects across the remaining twelve dimensions (Table 2). Fig 1 demonstrates participant responses to the vignettes.

### Immoral vignettes

In immoral vignettes, patients with schizophrenia rated other-benefit higher than control subjects ($F(1,62) = 9.7$, $p = 0.003$) and perceived the vignettes as less morally inappropriate ($F(1,62) = 15.5$, $p = 0.0002$). No significant difference was observed between patients with schizophrenia and control subjects in the remaining dimensions (Table 2, Fig 1).

### Other analyses

Analyses were performed to assess the effects of gender and disease duration (dichotomized by a median cutoff of 10 years) on provided responses, none of which showed a statistically significant association.

## Discussion

Our study evaluated moral decision-making in patients with schizophrenia compared to control subjects and found that patients with schizophrenia displayed altered moral decision-making. Patients rated moral appropriateness similarly to control subjects in moral vignettes, yet their perception of self-benefit for these vignettes was higher than

**Table 1. Demographic characteristics of participants.**

| Variable | Patients with schizophrenia (N=32) | Healthy subjects (N=32) | P value |
|---|---|---|---|
| Age | 38 (9.5) | 36.5 (9.5) | 0.652 |
| Gender (Male/Female) | 23/9 | 23/9 | 1 |
| Education level | 59.4% diploma | 40.6% diploma | 0.133 |
| | 40.6% bachelors | 59.4% bachelors | |
| Length of disease | 11 [1-39] | N/A | – |
| Number of hospitalization | 1.5 [0-15] | N/A | – |

Data presented as median (IQR), and median [range]

N/A: Not applicable

**Table 2. Differences between patients with schizophrenia and control subjects in moral and immoral vignettes.**

| Dimension | Moral Vignettes | | | | | Immoral Vignettes | | | | |
|---|---|---|---|---|---|---|---|---|---|---|
| | Patients responses* | Control subjects responses* | F test | P value | GES** | Patients responses* | Control subjects responses* | F test | P value | GES** |
| Emotional intensity | 4.68(1.42) | 4.23(1.11) | 1.93 | 0.17 | 0.03 | 4.17(1.21) | 3.78(1.22) | 3.857 | 0.054 | 0.026 |
| Emotional aversion | 3.76(1.04) | 3.32(1.02) | 2.89 | 0.094 | 0.045 | 4.7(0.86) | 4.39(1.03) | 3.405 | 0.07 | 0.04 |
| Harm | 2.86(1.18) | 2.48(0.78) | 2.165 | 0.146 | 0.034 | 4.56(0.74) | 4.80(0.54) | 0.718 | 0.4 | 0.029 |
| Self-benefit | 3.01(0.93) | 2.12[1-5] | 9.167 | 0.004 | 0.129 | 3.79(1.17) | 4.52[1.5-6] | 5.557 | 0.022 | 0.072 |
| Other-benefit | 4.09(0.92) | 3.81(1.21) | 1.09 | 0.301 | 0.017 | 2.69[1.22-4.47] | 1.98[1.12-3.28] | 9.7 | 0.003 | 0.114 |
| Pre-mediation | 3.03(1.25) | 3.06(1.04) | 0.007 | 0.935 | 0.00011 | 3.95(1.36) | 4.57(0.78) | 4.134 | 0.046 | 0.074 |
| Legality | 5.08(0.98) | 5.30(0.94) | 0.797 | 0.376 | 0.013 | 3.41(0.80) | 2.84[1.53-5.22] | 4.289 | 0.043 | 0.048 |
| Social norm violation | 5.08(0.98) | 4.74(0.82) | 2.205 | 0.143 | 0.034 | 3.12(0.87) | 2.23[1.06-5.94] | 4.204 | 0.045 | 0.06 |
| Socialness | 5.25(0.98) | 4.94[2.25-6.63] | 2.965 | 0.09 | 0.046 | 4.61(1.16) | 4.42(1.17) | 0.705 | 0.404 | 0.007 |
| Frequency | 4.14(1.25) | 4.28(0.71) | 0.342 | 0.561 | 0.005 | 3.98(1.42) | 4.40(1.18) | 1.512 | 0.224 | 0.027 |
| Personal familiarity | 3.02(1.17) | 3.49(0.83) | 3.435 | 0.069 | 0.052 | 2.016[1.12-4.5] | 1.97(0.46) | 3.694 | 0.059 | 0.065 |
| General familiarity | 4.03(1.28) | 4.48(0.73) | 2.974 | 0.09 | 0.046 | 4.07(1.37) | 4.45(0.93) | 1.151 | 0.288 | 0.027 |
| Moral appropriateness | 5.5(1.02) | 5.64(0.59) | 0.201 | 0.656 | 0.003 | 2.97(0.85) | 2.23(0.56) | 15.5 | 0.0002 | 0.214 |

Bonferroni correction: alpha = 0.0038

*Data presented as mean (SD) or median [range]

**Generalized Eta Squared

control subjects. In contrast, patients demonstrated distinct decision-making patterns in immoral vignettes, rating moral appropriateness higher than control subjects, considering a higher rate of other-benefit compared to control subjects.

In our study, patients with schizophrenia demonstrated a similar perception of moral appropriateness in moral vignettes compared to control subjects. This finding suggests that patients with schizophrenia have a comparable appreciation for prosocial situations to control subjects. However, patients attributed more self-gain in moral vignettes. Consistent with our findings, a study by Wischniewski et al. showed that patients with schizophrenia exhibit impaired strategic decision-making regarding self-gain [5]. It appears that reward-seeking behaviours in patients with schizophrenia may be influenced by an increased sensitivity to low motivational value stimuli [17]. The ceiling effect [18] might account for similar responses between patients and control subjects regarding moral appropriateness in moral vignettes, considering that all participants rated moral appropriateness at six out of a maximum of seven on the Likert scale for these vignettes.

We also found that patients with schizophrenia rated immoral vignettes as less morally inappropriate and perceived higher other-benefit for these vignettes compared to control subjects. Our findings can be explained by previous literature showing that patients with schizophrenia exhibit lower levels of empathy, particularly in the areas of empathic concern and perspective-taking [19]. Empathy deficiencies could explain why the patients conceive immoral vignettes as less inappropriate and assign higher benefits to such situations. Moreover, studies have shown that patients with schizophrenia score lower in the conscientiousness domain of personality traits compared to control subjects [20], which could affect patients' decision-making.

In our study, patients with schizophrenia exhibited difficulties in assessing benefits in both moral and immoral vignettes. The difficulties in assessing moral appropriateness and benefit in complex social situations could be related to various factors such as cognitive function [21], the theory of mind [22], and benefit-loss assessment, which are all known to be affected in patients with schizophrenia [4,5]. Cognitive processes play a pivotal role in moral decision-making and facilitate the evaluation of ethical dilemmas, integration of social and emotional considerations, and balancing interests. Impaired

**A**

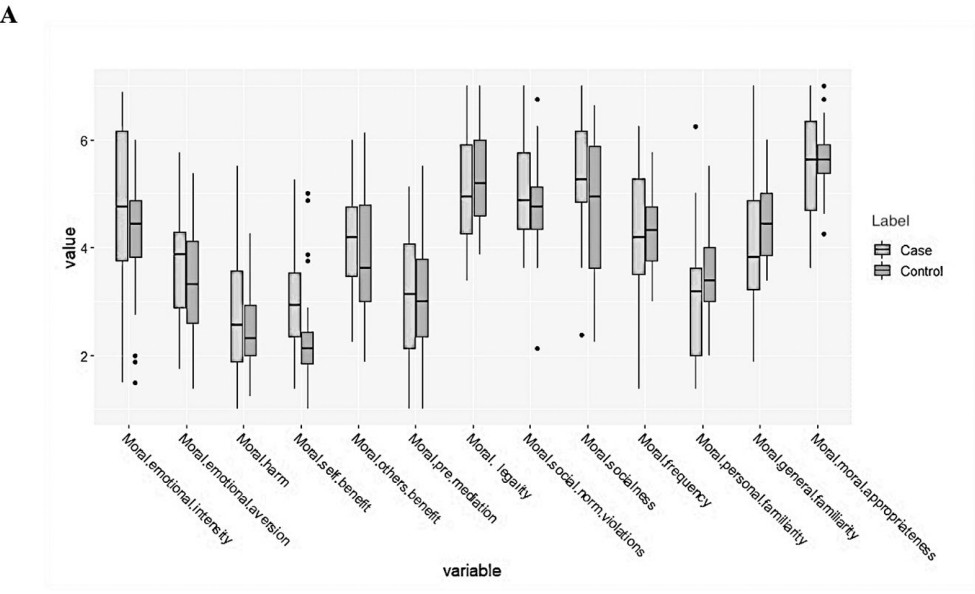

**B**

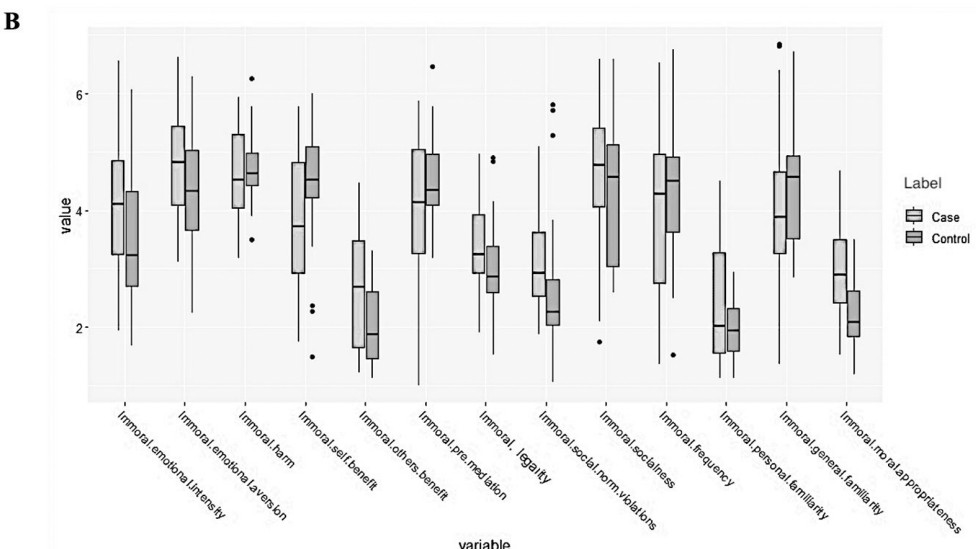

**Fig 1. Participants responses to the dimensions of moral (A) and immoral (B) vignettes.**

executive function, social cognition, and cognitive flexibility could significantly influence moral reasoning in patients [21,23,24]. Furthermore, theory of mind deficits, which affect the ability to understand others' perspectives and intentions, could mediate the observed differences in how patients and control subjects assess vignettes [21,22]. Future studies exploring both theory of mind and everyday moral decision-making in patients with schizophrenia could shed light on how the interaction of benefit and theory of mind could shape moral judgment. General intelligence has been argued to mediate moral decision-making [25] and affect sociomoral reasoning capabilities in patients with schizophrenia [26]. Nonetheless, it has been demonstrated that patients with preserved intellectual functioning have similar cognitive deficits to patients with compromised general intelligence, indicating that impaired cognitive processes are inherent characteristics

of schizophrenia regardless of intelligence levels [27]. Our study assessed educational degree as a measure of baseline cognitive ability and intelligence, and there were no significant differences among the groups.

We found no significant differences between the patients and control subjects in dimension of emotional intensity. This finding contrasts with previous research demonstrating differences between patients with schizophrenia and control subjects in the domain of emotion. A review of existing studies argues that the domain of emotion encompasses emotional expression, experience, and perception and recognition [28]. Considering the results and methodology of our study, it appears that the disparities between patients and control subjects in the domain of emotions are not universally applicable across all aspects of emotional processing and experience.

Furthermore, in our study, disease duration did not impact patient responses. While some studies have demonstrated that patients with chronic schizophrenia choose less favorable responses compared to those with first-episode psychosis [29], other studies have shown that most cognitive deficits of patients with schizophrenia are present from the initial episodes of psychosis. Additionally, research suggests that there is no significant decline in the deficits during the first ten years of the disease [30]. These findings align with our results as they indicate that disease duration may not be a determining factor in the differences observed between patients with schizophrenia and control subjects in moral decision-making.

The observed alterations in evaluating the benefit and moral appropriateness of actions among patients with schizophrenia could have significant implications for real-life social functioning. These changes could lead to deficient social cognition and difficulty interpreting and understanding others' intentions, resulting in strained personal and professional relationships [31,32]. Recent meta-analyses have reported that social-cognitive training, alongside the basic aspects of social cognition, can remediate more complex social-cognitive processes such as social perception and mental state attribution [12,33]. A combination of targeted interventions, such as cognitive behavioral therapy, social cognition and social skill training, and practice weighing self- and other-focused consequences in decisions, could improve perspective-taking and help restructure maladaptive thought patterns in patients [34,35].

## Limitations

One limitation of our study was the use of paper-and-pencil versions of moral vignettes, more immersive simulations could better replicate real-life situations. Although an accurate sample size calculation could not be made due to no available studies utilizing the same questionnaire in patients with schizophrenia, a conceivable sample size was calculated with 95% confidence level and a 5% margin of error as well as considering studies with comparable methodology [7,14,36,37]. However, low sample size of this study could affect the power of analysis to detect subtle differences between the two groups. Future research with additional sample sizes and implementation of additional measures such as Montreal Cognitive Assessment (MoCA), mini-mental state examination (MMSE), skin conductance response, and electroencephalograms (EEG) could provide further insights into the underlying mechanisms of moral decision-making in patients with schizophrenia.

## Conclusion

Our study suggests that most aspects of everyday moral decision-making remain intact in patients with schizophrenia. The observed deficits appear to be limited to the evaluation of moral appropriateness in immoral situations and overall benefit assessment. The distortion in benefit assessment for oneself and others could be a crucial area of focus for cognitive rehabilitation of patients with schizophrenia. By addressing this issue we may help improve moral decision-making abilities in everyday life situations.

## Acknowledgments

None.

## Author contributions

**Conceptualization:** Fatemeh Sadat Mirfazeli, Seyed Vahid Shariat.

**Data curation:** Koohyar Ahmadzadeh.

**Formal analysis:** Atiye Sarabi-Jamab.

**Investigation:** Koohyar Ahmadzadeh, Fatemeh Sadat Mirfazeli, Seyed Vahid Shariat.

**Methodology:** Fatemeh Sadat Mirfazeli, Seyed Vahid Shariat.

**Project administration:** Seyed Vahid Shariat.

**Writing – original draft:** Koohyar Ahmadzadeh, Atiye Sarabi-Jamab, Farahnaz Beiranvand, Mina Forouzandeh, Fatemeh Sadat Mirfazeli, Seyed Vahid Shariat.

**Writing – review & editing:** Koohyar Ahmadzadeh, Atiye Sarabi-Jamab, Farahnaz Beiranvand, Mina Forouzandeh, Fatemeh Sadat Mirfazeli, Seyed Vahid Shariat.

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
