## [Decision Letter · Decision Letter 0]

3 Jan 2025

PONE-D-24-51462Moral decision-making is altered in patients with schizophreniaPLOS ONE

Dear Dr. Shariat,

Thank you for submitting your manuscript to PLOS ONE. After careful consideration, we feel that it has merit but does not fully meet PLOS ONE’s publication criteria as it currently stands. Therefore, we invite you to submit a revised version of the manuscript that addresses the points raised during the review process.

We look forward to receiving your revised manuscript.

Kind regards,

Lakshminarayana Chekuri, MD, PhD

Academic Editor

PLOS ONE

2. Please describe in your methods section how capacity to provide consent was determined for the participants in this study. Please also state whether your ethics committee or IRB approved this consent procedure. If you did not assess capacity to consent please briefly outline why this was not necessary in this case.

3. In the online submission form you indicate that your data is not available for proprietary reasons and have provided a contact point for accessing this data. Please note that your current contact point is a co-author on this manuscript. According to our Data Policy, the contact point must not be an author on the manuscript and must be an institutional contact, ideally not an individual. Please revise your data statement to a non-author institutional point of contact, such as a data access or ethics committee, and send this to us via return email. Please also include contact information for the third party organization, and please include the full citation of where the data can be found.

5. We note that you have referenced (Escobedo JR. Investigating moral events: characterization and structure of autobiographical moral memories. (Unpublished dissertation), Pasadena, CA: California Institute of Technology. 2009.) which has currently not yet been accepted for publication. Please remove this from your References and amend this to state in the body of your manuscript: (ie “Bewick et al. [Unpublished]”) as detailed online in our guide for authors http://journals.plos.org/plosone/s/submission-guidelines#loc-reference-style

Additional Editor Comments:

Please see attached document for my comments.

Reviewers' comments:

Reviewer's Responses to Questions

**Comments to the Author**

1. Is the manuscript technically sound, and do the data support the conclusions?

Reviewer #1: Yes

Reviewer #2: Yes

2. Has the statistical analysis been performed appropriately and rigorously? 

Reviewer #1: Yes

Reviewer #2: Yes

3. Have the authors made all data underlying the findings in their manuscript fully available?

Reviewer #1: Yes

Reviewer #2: No

4. Is the manuscript presented in an intelligible fashion and written in standard English?

Reviewer #1: Yes

Reviewer #2: Yes

5. Review Comments to the Author

Reviewer #1: Thank you for the opportunity to review the manuscript "Moral decision-making is altered in patients with schizophrenia." This is an important contribution to understanding moral decision-making processes in schizophrenia and has potential implications for improving interventions targeting social functioning. While the study is well-conducted, some refinements would strengthen its impact:

1. Statistical Presentation:

- Table 1 would benefit from including descriptive statistics (means, standard deviations) for both groups

- Addition of effect sizes and confidence intervals would help readers better understand the magnitude of differences between groups

2. Intervention Context:

- The discussion would benefit from better contextualizing the findings within the broader framework of interventions mentioned in the introduction

- Further elaboration on how the observed alterations in moral decision-making relate to real-life social functioning challenges

- Clearer implications of these findings for developing targeted interventions to enhance social functioning in patients with schizophrenia

3. Intelligence and Cognitive Function:

- A more detailed discussion of how cognitive function might influence moral decision-making

- Clarification of whether and how intelligence was assessed beyond basic screening

- Discussion of the relationship between general intelligence and moral reasoning capacity

- How cognitive function might mediate the observed differences between groups

Reviewer #2: This paper highlights the difficulties encountered by patients with schizophrenia in judging everyday moral vignettes compared to healthy peers. The paper requires some minor revisions that will enhance its formal quality.

Minor Comments:

1. According to the journal’s submission guidelines, the introduction should include a brief statement of the overall aim of the work, along with a comment on whether that aim was achieved.

2. In the description of the task, it is not explained how the individual scores were measured. Was this a median score computed from the individual's responses?

3. The description of the statistical analysis needs revision, as it is currently not clear.

4. Demographic characteristics (such as gender and education) are not described. Please add this information to the text or present it in a table.

5. In Table 1, I recommend presenting the F-values first, followed by the p-values.

6. PLOS authors have the option to publish the peer review history of their article (what does this mean? ). If published, this will include your full peer review and any attached files.

**Do you want your identity to be public for this peer review?** For information about this choice, including consent withdrawal, please see our Privacy Policy .

Reviewer #1: **Yes: ** Alessandro Rodolico

Reviewer #2: No

---

## [Author Response · Author response to Decision Letter 1]

8 Feb 2025

We greatly thank the respected editors and reviewers for their valuable time and comments on our manuscript. We have thoroughly considered their constructive comments and amended the manuscript accordingly.

In short, discussion and last paragraph of introduction have been amended according to the comments of respected editors and reviewers. The discussion now contains sections on the interplay of cognition and moral reasoning, and functional implications and interventions. Methods section has been edited to more clearly present patient inclusion and analysis rationale. Psychometric properties of utilized questionnaires have also been added. Lastly, a new table containing patient demographics has been added to the results, and the previous table has been amended to contain more detailed presentations of results. The respective change regarding each point is tracked in the uploaded revised manuscript.

We’d also like to note that the affiliation for “Atiye Sarabi-Jamab” and “Fatemeh Sadat Mirfazeli” have been updated. We apologize for any inconvenience this might cause. We would like to mention that Fatemeh Sadat Mirfazeli is the Co-corresponding author with Dr. Seyed Vahid Shariat.

Thank you very much for your assistance in enhancing our manuscript. Below are our point-to-point responses to the comments:

Academic Editor Comments:

1. On page 3, in the materials and methods "Participants" section, please provide inclusion criteria for the study participants.

Author response:

Thank you for your comment to enhance the quality of the manuscript. The methods section regarding participant inclusion has been rewritten to present the process more clearly for patients and control subjects.

Manuscript has been changed as follows:

“This cross-sectional study recruited 64 participants (32 patients, 32 control subjects) from clinics of university-affiliated hospitals between June 1st and November 30th 2021.

Patients from two different outpatient psychiatry clinics who were diagnosed with schizophrenia according to the Diagnostic and Statistical Manual of Mental Disorders (DSM)-5, as established by an attending psychiatrist, were included in the study.”

“Control subjects were recruited from all outpatient clinics of university-affiliated hospitals, excluding psychiatry, psychology, neurology, and neurosurgery clinics. Subjects with a General Health Questionnaire (GHQ-12) score of less than 23 were included as the control group.”

2. On page 4, under "measures" section, Please provide psychometric properties for "the moral questionnaire" and for the "GHQ-12".

Author response:

Thank you for your comment. The methods section now contains the measures for reliability of both questionnaires.

“Due to the length of the original questionnaire, a shorter version, containing 40 vignettes, has been translated, abbreviated, and validated in the Persian language by Yazdanpanah et al [14], which was utilized in this study. The internal consistency of the shortened vignettes in the Iranian culture, measured by Cronbach’s alpha, was 0.895. The Cronbach’s alpha measurement of the original 150-vignette questionnaire was 0.96 and it was shown to be consistent with Iranian culture [14].”

“The mental health of control subjects was measured by a Persian-translated and validated version of GHQ-12. This questionnaire is considered valid and reliable across various settings and cultures, such as Iran (Cronbach’s alpha measure: 0.87, Convergent validity: r = -.56, P<0.0001) [15], Finland (Cronbach’s alpha measure: 0.92), India (Cronbach’s alpha measure: 0.82), and South Korea (Cronbach’s alpha measure: 0.81) [16].”

3. On page 5, under "Demographic characteristics", please consider changing "overall mean age" to "overall mean age in years"

Author response:

Thank you for your comment. A change has been made to the sentence accordingly.

4. On page 5, under "Demographic characteristics", please clarify why the standard deviation was greater than the mean or is it a typo? ".... with 2.16 ± 2.92 numbers of hospitalization."

Author response:

Thank you for your meticulousness. Due to the data's non-normality, the SD was greater than the mean. To better represent the baseline characteristics, the data presentation has been changed to median for age and median [range] for disease duration and number of hospitalizations in the manuscript text.

5. Please provide if there were any dropouts or if any of the items had a non-response:

Author response:

Thank you for your comment. There were no dropouts throughout the study, and participants responded to all items in the questionnaire. This information is now included in the methods section.

6. One of the limitations of this study could be a relatively low sample size which could have affected the power to detect subtle differences between the two groups. If you agree with this assessment please acknowledge this in the limitations section on page 8.

Author response:

The sample size calculations were based on previous publications with comparable methodologies [7, 14, 36, 37] and the below formula, considering 95% confidence level and a 5% margin of error which counts a sample size of around 64, 32 control groups, and 32 patients.

n_1=〖(σ_1^2+(σ_2^2)⁄K)(Z_(1-α⁄2)+Z_(1-β))〗^2/∆^2

k=n_2/n_1 =1

However, the relatively low sample size could have affected the power of our analysis, and studies with larger sample sizes could detect any other possible subtle differences between the two groups. This limitation is mentioned in the limitations section of the manuscript.

Manuscript has been changed as follows:

“Although an accurate sample size calculation could not be made due to no available studies utilizing the same questionnaire in patients with schizophrenia, a conceivable sample size was calculated with 95% confidence level and a 5% margin of error as well as considering studies with comparable methodology [7, 14, 36, 37]. However, low sample size of this study could affect the power of analysis to detect subtle differences between the two groups.”

7. On page 3, the authors state "Moreover, as the purpose of our study, the focus on everyday moral dilemmas will provide precious insights into the functional implications of these deficits, ultimately contributing to the development of targeted interventions to enhance social functioning in patients with schizophrenia." However, I could not see any reasonable discussion on "insights into the functional implications of these deficits" in the discussion section. Please address this in your next revision.

Author response:

Thank you for your comment. The discussion has been amended to further explain the possible functional implications and targeted interventions. The mentioned sentences of the introduction have also been slightly adjusted.

Manuscript has been changed as follows:

Introduction: “Moreover, the focus on everyday moral dilemmas with multi-aspect dimensions could provide precious insights into the functional implications of these deficits, ultimately contributing to the development of targeted interventions such as socio-cognitive rehabilitation to enhance social functioning in patients with schizophrenia [12].”

Discussion: “The observed alterations in evaluating the benefit and moral appropriateness of actions among patients with schizophrenia could have significant implications for real-life social functioning. These changes could lead to deficient social cognition and difficulty interpreting and understanding others intentions, resulting in strained personal and professional relationships [31, 32]. Recent meta-analyses have reported that social-cognitive training, alongside the basic aspects of social cognition, can remediate more complex social-cognitive processes such as social perception and mental state attribution [12, 33]. A combination of targeted interventions, such as cognitive behavioral therapy, social cognition and social skill training, and practice weighing self- and other-focused consequences in decisions, could improve perspective-taking and help restructure maladaptive thought patterns in patients [34, 35].”

1. Please ensure that your manuscript meets PLOS ONE's style requirements.

Author response:

The manuscript has been edited according to journal requirements.

2. Please describe in your methods section how capacity to provide consent was determined for the participants in this study. Please also state whether your ethics committee or IRB approved this consent procedure. If you did not assess capacity to consent please briefly outline why this was not necessary in this case.

Author response:

Thank you for your comment. The methods section, under “ethical considerations”, now explains how capacity to provide consent was determined.

Additionally, the ethics committee of IUMS approved this procedure and all the steps of the study.

Manuscript has been changed as follows:

“The capacity to provide consent was determined through the clinical judgment of the attending physician, ensuring participants understood the study’s purpose and their voluntary participation. The patients who had the capacity for consent were included in the study. The ethics committee of the Iran University of Medical Sciences provided approval for all study procedures.”

3. In the online submission form you indicate that your data is not available for proprietary reasons and have provided a contact point for accessing this data. Please note that your current contact point is a co-author on this manuscript. According to our Data Policy, the contact point must not be an author on the manuscript and must be an institutional contact, ideally not an individual. Please revise your data statement to a non-author institutional point of contact, such as a data access or ethics committee, and send this to us via return email. Please also include contact information for the third party organization, and please include the full citation of where the data can be found.

Author response:

The contact point for accessing data has been changed to the National Brain Centre and the manuscript has been adjusted accordingly.

Author response:

The ORCID iD for the corresponding authors has been added to the profile page in the editorial manager.

5. We note that you have referenced (Escobedo JR. Investigating moral events: characterization and structure of autobiographical moral memories. (Unpublished dissertation), Pasadena, CA: California Institute of Technology. 2009.) which has currently not yet been accepted for publication. Please remove this from your References and amend this to state in the body of your manuscript: (ie “Bewick et al. [Unpublished]”) as detailed online in our guide for authors http://journals.plos.org/plosone/s/submission-guidelines#loc-reference-style

Author response:

Thank you for your meticulousness. Considering that the mentioned “Escobedo JR et. al.” manuscript is a published dissertation with an assigned DOI, this citation is now edited and entered as a “Master's thesis or doctoral dissertation” according to the instructions provided in the PLOS ONE journal webpage.

Additionally, the manuscript now uses the PLoS referencing style suggested in the journal webpage.

Review Comments to the Author:

Reviewer #1:

We greatly thank the respected reviewers for their valuable time and comments on our manuscript. We have thoroughly considered their constructive comments and amended the manuscript accordingly.

1. Statistical Presentation:

- Table 1 would benefit from including descriptive statistics (means, standard deviations) for both groups.

- Addition of effect sizes and confidence intervals would help readers better understand the magnitude of differences between groups

Author response:

Thank you for your valuable comments. In line with adding a new table, previous table one has been updated to table two and now contains descriptive statistics (mean/SD and median/range) for responses of both groups. In addition, the Generalized Ets Squared (GES) values have been added to the table, as a representative of effect size.

Considering the already crowded table 2, we refrained from reporting additional values listed in the table below, including the difference between control subjects minus patients with 95% CI. These values could be reported in the manuscript if the respected reviewer deems it necessary.

[Please view the uploaded response to reviewers word file for a correct presentation of tables]

Dimension Moral Vignettes Immoral Vignettes

generalized eta squared (ges) diff(control subjects – patients with schizophrenia) 95% confidence interval* p-adj generalized eta squared (ges) diff(control subjects - patients with schizophrenia) 95% confidence interval* p-adj

Emotional intensity 0.03 -0.44 -1.08, 0.19 0.17 0.026 -0.39 -1.00

0.22 0.21

Emotional aversion 0.045 -0.43 -0.95, 0.08 0.94 0.04 -0.38 -0.85

0.09 0.11

Harm 0.034 -0.37 -0.87, 0.13 0.15 0.029 0.22 -0.10, 0.55 0.18

Self-benefit 0.129 -0.70 -1.17, -0.28 0.004 0.072 0.59 0.05, 1.13 0.03

Other benefit 0.017 -0.28 -0.82, 0.28 0.30 0.0114 -0.62 -1.07, -0.18 0.00

Pre-mediation 0.00011 0.02 -0.55, 0.59 0.94 0.074 0.62 0.06, 1.17 0.03

Legality 0.013 0.21 -0.27, 0.70 0.38 0.048 -0.36 -0.76, 0.05 0.08

Social norm violation 0.034 -0.34 -0.79, 0.12 0.14 0.06 -0.51 -1.02, 0.00 0.05

Socialness 0.046 -0.53 -1.13, 0.08 0.09 0.007 -0.19 -0.77, 0.39 0.51

Frequency 0.005 0.15 -0.39, 0.66 0.56 0.027 0.42 -0.23, 1.07 0.19

Personal familiarity 0.052 0.47 -0.04, 0.98 0.07 0.065 -0.41 -0.81, -0.01 0.04

General familiarity 0.046 0.45 -0.07, 0.97 0.09 0.027 0.38 -0.20, 0.96 0.19

Moral appropriateness 0.003 0.09 -0.33, 0.51 0.65 0.214 -0.74 -1.09, -0.38 0.00

2. Intervention Context:

- The discussion would benefit from better contextualizing the findings within the broader framework of interventions mentioned in the introduction

- Further elaboration on how the observed alterations in moral decision-making relate to real-life social functioning challenges

- Clearer implications of these findings for developing targeted interventions to enhance social functioning in patients with schizophrenia:

Author response:

Thank you for your precious comment on enhancing the quality of our work. A paragraph has been added to the discussion to elaborate on the context of real-life functioning and implications for interventions.

Manuscript has been changed as follows:

“The observed alterations in evaluating the benefit and moral appropriateness of actions among patients with schizophrenia could have significant implications for real-life social functioning. These changes could lead to deficient social cognition and difficulty interpreting and understanding others intentions, resulting in strained personal and professional relationships [31, 32]. Recent meta-analyses have reported that social-cognitive training, alongside the basic aspects of social cognition, can remediate more complex social-cognitive processes such as social perception and mental state attribution [12, 33]. A combination of targeted interventions, such as cognitive behavioral therapy, social cognition and social skill training, and practice weighing self- and other-focused consequences in decisions, could improve perspective-taking and help restructure maladaptive thought patterns in patients [34, 35].”

3. Intelligence and Cognitive Function:

- A more detailed discussion of how cognitive function might influence moral decision-making

- Clarification of whether and how intelligence was assessed beyond basic screening: methods or limitation or discussion?

- Discussion of the relationship between general intelligence and moral reasoning capacity

- How cognitive function might mediate the observed diffe

---

## [Decision Letter · Decision Letter 1]

11 Mar 2025

PONE-D-24-51462R1Moral decision-making is altered in patients with schizophreniaPLOS ONE

Dear Dr. Shariat,

Thank you for submitting your manuscript to PLOS ONE. After careful consideration, we feel that it has merit but does not fully meet PLOS ONE’s publication criteria as it currently stands. Therefore, we invite you to submit a revised version of the manuscript that addresses the points raised during the review process.

Please review and address comments in attached document.

We look forward to receiving your revised manuscript.

Kind regards,

Lakshminarayana Chekuri, MD, PhD

Academic Editor

PLOS ONE

Journal Requirements:

Additional Editor Comments :

Please review and address comments in attached document.

Reviewers' comments:

Reviewer's Responses to Questions

**Comments to the Author**

1. If the authors have adequately addressed your comments raised in a previous round of review and you feel that this manuscript is now acceptable for publication, you may indicate that here to bypass the “Comments to the Author” section, enter your conflict of interest statement in the “Confidential to Editor” section, and submit your "Accept" recommendation.

Reviewer #2: All comments have been addressed

2. Is the manuscript technically sound, and do the data support the conclusions?

Reviewer #2: Yes

3. Has the statistical analysis been performed appropriately and rigorously? 

Reviewer #2: Yes

4. Have the authors made all data underlying the findings in their manuscript fully available?

Reviewer #2: Yes

5. Is the manuscript presented in an intelligible fashion and written in standard English?

Reviewer #2: Yes

6. Review Comments to the Author

Reviewer #2: (No Response)

7. PLOS authors have the option to publish the peer review history of their article (what does this mean? ). If published, this will include your full peer review and any attached files.

**Do you want your identity to be public for this peer review?** For information about this choice, including consent withdrawal, please see our Privacy Policy .

Reviewer #2: No

---

## [Author Response · Author response to Decision Letter 2]

17 Mar 2025

Thank you very much for taking the time to revise and improve our manuscript. Below are our responses to your valuable comments:

Academic Editor Comments:

1. Abstract: Results section: Please provide p values in parenthesis at the end of first two statements.

Author response:

Thank you for your comment. Abstract now contains the requested p values.

2. Line 89: Participants: Did you consider age of participants while recruiting study participants? If so mention this in the inclusion / exclusion criteria.

Author response:

Thank you for your comment. Participants younger than 18 and older than 65 years were excluded from the study. This information is now mentioned in line 103.

3. Line 104: What was the criteria used to define "alcohol addiction" and how did you screen participants for "alcohol addiction". Please explain and include in this section.

Author response:

Thank you for your comment. Alcohol use in patients was judged based on DSM-5 criteria for alcohol use disorder. “Alcohol addiction” has been altered to “alcohol use disorder” for more precise definition.

The manuscript now mentions that “Alcohol use disorder was assessed through interview based on DSM-5 criteria.”.

Journal Requirements:

Author response:

The reference list was reviewed to make sure all references match journal requirements. The PLoS style has been used for reference management.

The data availability statement has also been edited to explain the restrictions further, as requested by the respected editorial office.

We once again thank the editorial team for their attentive review of our manuscript and are at your disposal for any further changes that may be required.

---

## [Editor Report · Decision Letter 2]

25 Mar 2025

Moral decision-making is altered in patients with schizophrenia

PONE-D-24-51462R2

Dear Dr. Shariat,

We’re pleased to inform you that your manuscript has been judged scientifically suitable for publication and will be formally accepted for publication once it meets all outstanding technical requirements.

Kind regards,

Lakshminarayana Chekuri, MD, PhD

Academic Editor

PLOS ONE
---

## [Editor Report · Acceptance letter]

PONE-D-24-51462R2

PLOS ONE

Dear Dr. Shariat,

I'm pleased to inform you that your manuscript has been deemed suitable for publication in PLOS ONE. Congratulations! Your manuscript is now being handed over to our production team.

Kind regards,

on behalf of

Dr. Lakshminarayana Chekuri

Academic Editor

PLOS ONE